# OpenReview forum: "PrefDisco: Benchmarking Proactive Personalized Reasoning"
_ICLR.cc/2026/Conference — ICLR 2026 Poster_

### Official Review · Reviewer_zi3F · 2025-10-29

**Soundness:** 2
**Presentation:** 2
**Contribution:** 1
**Rating:** 2
**Confidence:** 3

**Summary:**

The paper introduces personalized reasoning, the ability of large language models (LLMs) to adapt their reasoning and explanations to individual user preferences rather than offering one-size-fits-all answers. It presents PREFDISCO, a benchmark that converts static reasoning tasks into interactive personalization challenges where models must elicit user preferences through questioning and tailor responses accordingly. Evaluating 21 frontier models across 10 domains, the authors find that 29% of personalization attempts reduce alignment compared to generic responses and often lower task accuracy, especially in mathematical reasoning. The study concludes that effective personalization requires dedicated mechanisms for preference discovery and adaptive reasoning, as current LLMs fail to perform just-in-time personalization despite strong general capabilities.

**Strengths:**

They have assessed the 21 frontier models across multiple benchmarks.

**Weaknesses:**

I believe addressing the following concern might help to improve the quality:

1. Although the task is interesting, the exploration of the user preference has already been conducted in UserBench [1]. The paper lacks a comprehensive comparison with existing works. No table or section is provided for such comparisons. I encourage the authors to conduct a thorough literature review and include a detailed comparative analysis with current studies.


2. Another concern relates to the five interaction turns used in the experiments. The rationale behind selecting this specific number is not explained. An ablation study exploring the impact of different conversation counts would be helpful to justify this choice.


3. I believe that preference reasoning presents a valuable direction to explore for planning-related tasks, such as travel planning. I suggest that the authors include this domain in the benchmark to enhance the coverage and applicability of the study.

4. The process used to generate user instructions is not clearly described. Only one incomplete example is presented in Figure 1. It is recommended that the authors release a portion of the benchmark and provide a detailed visualization of the data generation process.


5. The authors mention that the user simulator is passive. However, in practice, an assistant faced with a passive user would likely disengage and terminate the conversation. The reasoning for considering a passive user in the experiments remains unclear and requires further justification.


6. The authors state that MATH and LogiQA exhibit the most performance degradation, whereas SocialIQA benefits the most from interactive personalization. However, no analysis or explanation is provided to clarify these observations. A deeper analysis would help readers better understand the underlying performance dynamics.


7. The results in Table 1 show that, for most benchmarks, the performance of 3-Opus surpasses that of Sonnet-4. However, Sonnet-4 is a stronger model. The authors should provide an explanation for this phenomenon.


8. The judgment process used in the evaluation is insufficiently explained. Including the evaluation prompts and methodology details would enhance transparency and reproducibility.


---
**Reference**

[1] UserBench: https://arxiv.org/pdf/2507.22034

**Questions:**

Please read the weaknesses section.

---

> ### Author Response · Authors · 2025-11-21
>
> We thank the reviewer for acknowledging that we assessed 21 frontier models across multiple benchmarks.
>
> ## Weakness 1
>
> > "Although the task is interesting, the exploration of the user preference has already been conducted in UserBench. The paper lacks a comprehensive comparison with existing works."
>
> We thank the reviewer for pointing out UserBench. While both UserBench and PREFDISCO involve eliciting user preferences, these benchmarks evaluate fundamentally different problems. UserBench evaluates tool-using planning agents in a single domain (travel), where preferences concern external action choices (e.g., flights/hotels). PREFDISCO instead studies personalized reasoning: how models elicit fine-grained explanation preferences (e.g., detail, empathy, jargon level) and adapt their reasoning process and explanations across ten diverse reasoning domains. Below we provide a detailed comparison table:
>
> | Aspect | UserBench | PREFDISCO |
> |--------|-----------|-----------|
> | Core problem | Tool-using agents in multi-turn travel-planning | Personalized reasoning with explanation adaptation |
> | Task setting | Single domain: structured travel planning | 10 diverse reasoning domains (math, science, medicine, etc.) |
> | Interaction goal | Identify constraints and choose travel options | Elicit explanation preferences and tailor reasoning accordingly |
> | Personalization target | Outcome/action selection (which flight/hotel) | Reasoning process + explanation style while maintaining correctness |
> | Preference modeling | Fixed travel-specific preferences expressed implicitly | Sparse, instance-level profiles with values and importance weights |
> | User model | Simulated user in Gym environment | Psychologically grounded personas with explicit elicitation |
> | Evaluation | Correctness of chosen travel actions | Task correctness + fine-grained preference alignment |
> | Tools/environment | Heavy use of tool APIs, noise, cost-aware interactions | No tool use; focus on reasoning and explanation adaptation |
>
> While travel planning has been studied in UserBench, our framework is extendible to this domain. However, since our focus is on reasoning personalization rather than tool-based action selection, we prioritized coverage across diverse reasoning tasks.
>
> ## Weakness 2
>
> > "Another concern relates to the five interaction turns used in the experiments. The rationale behind selecting this specific number is not explained. An ablation study exploring the impact of different conversation counts would be helpful"
>
> We explicitly study the effect of varying the number of questions in Lines 428–439 and Figure 5. Our goal is to isolate question quality from termination decisions, so we force models to ask a fixed number of questions (2, 4, or 8) rather than allowing them to decide when to stop.
>
> This controlled setting reveals that domain-specific performance patterns persist regardless of question quantity. Mathematical and scientific reasoning tasks (MATH, MascQA) continue to degrade as more questions are asked, whereas social reasoning tasks (CommonsenseQA, SocialIQA) consistently improve. Because performance curves saturate around 3–5 turns, and additional turns mostly introduce cognitive overhead without improving alignment, we selected five turns as a balanced, empirically supported operating point. We will make this rationale clearer in the revised manuscript.
>
> ## Weakness 3
>
> > "I believe that preference reasoning presents a valuable direction to explore for planning-related tasks, such as travel planning. I suggest that the authors include this domain in the benchmark"
>
> We agree that personalization is valuable for travel planning. As shown in our comparison with UserBench, our framework can be extended to travel planning by extracting instance-specific explanation or decision attributes (e.g., risk tolerance, comfort vs. cost trade-offs). However, our work targets a fundamentally different goal (personalized reasoning and explanation adaptation while solving a *verifiable* problem correctly rather than tool-based action selection), we focused on diverse reasoning domains in the initial submission. We will incorporate a travel-planning personalization variant in future work to broaden coverage while preserving the distinction between reasoning personalization and tool-based decision-making.
>
> ## Weakness 4
>
> > "The process used to generate user instructions is not clearly described. It is recommended that the authors release a portion of the benchmark and provide a detailed visualization of the data generation process."
>
> As we state in the paper, the benchmark is **entirely** publicly available, we cannot provide the link to the benchmark due to anonymity, but we have added multiple instances of the benchmark and complete interaction and evaluation traces in the appendix. Figure 1 provides a detailed visualization of the data generation process and our generation pipeline will also be publicly released.

---

> > ### Author Response · Authors · 2025-11-21
> >
> > ## Weakness 5
> >
> > > "The authors mention that the user simulator is passive. However, in practice, an assistant faced with a passive user would likely disengage and terminate the conversation. The reasoning for considering a passive user in the experiments remains unclear"
> >
> > Research on human-AI interaction shows that users frequently exhibit passive behavior when interacting with AI systems. Studies on interactive AI systems have found that users often adopt a passive role where they provide minimal information unless explicitly prompted, as opposed to actively engaging in co-creation with AI [1-2]. The passive acceptance of AI outputs without comprehensive understanding of system capabilities is a well-documented phenomenon in human-AI interaction research [3].
> >
> > Using a passive simulator lets us directly test whether models can proactively identify missing information, ask the right questions, and adapt their reasoning without relying on user initiative. This isolates elicitation skill rather than conflating it with a cooperative user.
> >
> > In response to the reviewer's concern, we have added a collaborative user scenario, where users provide textual descriptions and proactively reveal preferences without being prompted, addressing the reviewer's concern about realism. We will report these results in the camera-ready version. Additionally, the system prompt instructs the assistant to ask questions regardless of user interactivity level until the user explicitly terminates the conversation. We will add the complete system prompts to the appendix for transparency.
> >
> > ## Weakness 6
> >
> > > "The authors state that MATH and LogiQA exhibit the most performance degradation, whereas SocialIQA benefits the most from interactive personalization. However, no analysis or explanation is provided"
> >
> > We provide analysis in Lines 410–427 and Figure 5. Mathematical reasoning tasks require precise formal logic where additional contextual constraints (preference requirements) interfere with the optimization patterns models learned during training. We are currently conducting detailed analysis of interaction traces showing specific cases where mathematical reasoning degrades under personalization constraints versus social reasoning tasks that benefit from additional context. These qualitative examples will be included in the camera-ready version. Additionally, prior work has shown that slight rewording of math problems can cause significant degradation in the model reasoning and explore why this happens [4], although out of scope for our paper.
> >
> >
> > ## Weakness 7
> >
> > > "The results in Table 1 show that, for most benchmarks, the performance of 3-Opus surpasses that of Sonnet-4. However, Sonnet-4 is a stronger model."
> >
> > This observation highlights an important property of our evaluation. While Sonnet-4 belongs to a newer model generation and is generally stronger on conventional task-only benchmarks, this does not imply superiority on the interactive personalization abilities that PREFDISCO measures. Static benchmarks primarily reward single-turn problem solving, and recent models may be increasingly optimized for these settings, potentially overfitting to narrow solution patterns.
> >
> > Recent work shows that alignment techniques like RLHF and RLAIF reduce models' conceptual diversity compared to their base or instruction fine-tuned counterparts [5]. Less aligned models often exhibit more diversity in their outputs, making them potentially more adaptable to personalized tasks. PREFDISCO reveals this discrepancy: 29% of all model-task combinations yield negative discovery-mode scores, indicating that attempted personalization can actively harm alignment. The instances where 3-Opus surpasses Sonnet-4 reinforce our broader takeaway: gains in task accuracy do not necessarily translate into gains in preference elicitation or adaptive dialogue. This represents a unique capability that is not being optimized for in current benchmark-driven evaluation practices.
> >
> >
> > ## Weakness 8
> >
> > > "The judgment process used in the evaluation is insufficiently explained. Including the evaluation prompts and methodology details would enhance transparency"
> >
> > Thank you for raising the confusion. We will add complete evaluation prompts and methodology details to the appendix, including the rubric-based evaluation protocol and example judge outputs with justifications.
> >
> >
> > Overall, we thank the reviewer for their feedback and hope that our response has addressed the reviewer's concerns and reduced any potential misunderstanding. We would appreciate if the reviewer could engage in the discussion process and raise scores accordingly if their questions have been addressed.

---

> > > ### Author Response · Authors · 2025-11-21
> > >
> > > ## References
> > >
> > > [1] Towards algorithmic experience: Initial efforts for social media contexts (Alvarado & Waern, CHI 2018)
> > > [2] Rise of machine agency: A framework for studying the psychology of human–AI interaction (HAII). (Sundar, S. S., Journal of computer-mediated communication, 2020)
> > > [3] Beyond the dichotomy of use and not-use: Forms and motivations of user non-use behaviors toward AI customer service. Computers in Human Behavior. (Yao et al., Computers in Human Behavior, 2025)
> > > [4] Spurious Rewards: Rethinking Training Signals in RLVR (Shao et al., 2025)
> > > [5] One fish, two fish, but not the whole sea: Alignment reduces language models’ conceptual diversity (Murthy et al., NAACL 2025)

---

### Official Review · Reviewer_9SvA · 2025-10-29

**Soundness:** 3
**Presentation:** 2
**Contribution:** 2
**Rating:** 4
**Confidence:** 3

**Summary:**

This paper defines personalized reasoning in just-in-time scenarios, where the model actively seeks answers to discover user preferences and adjusts its reasoning accordingly. It then proposes PREFDISCO, transforming a static benchmark into an interactive task using psychology-driven sparse preference profiles. Experimental results demonstrate the limitations of current LLMs in personalized reasoning.

**Strengths:**

1. Extending from personalized content to personalized reasoning is interesting and reasonable.

2. A psychology-driven evaluation method is proposed.

3. Experimental results highlight the limitations of current LLMs, emphasizing the need to consider this characteristic.

**Weaknesses:**

1. Many details in the proposed evaluation framework remain unclear. For example, how are sparse preference profiles generated? What are the 20 scenarios? What are the relevant and irrelevant attributes for each scenario? How are importance weights calculated? How is LLM-based assessment performed? The numerous vague descriptions make the paper incomplete.

2. While passive user simulation can simulate more challenging scenarios, it may overlook more informative interactions in general. Furthermore, real-world scenarios are not always extreme and may not accurately reflect actual situations. Therefore, this rationale requires further discussion.

3. The paper's analysis is not sufficiently thorough. For example, the paper attributes the significant drop in accuracy of the mathematical task to "over-optimization," but can this be further demonstrated experimentally? Why does the accuracy of the mathematical task continue to decline with increasing question count?

4. Case studies would be beneficial. For example, showcasing various failure cases to support the paper's arguments.

5. Minor: The range of the value of $g_j(r,v_j)$ is inconsistent between Sections 2.3 and 3.

**Questions:**

Please refer to Weaknesses.

---

> ### Author Response · Authors · 2025-11-25
>
> We thank the reviewer for acknowledging our novelty in proposing a new capability of personalized reasoning, and demonstrating across many domains how this is a limitation of the current frontier LLMs.
>
> **Weakness 1**
>
>  > “How are sparse preference profiles generated?”
>
> We refer the reviewer to Section F in our appendix where we have included all the prompts we used to generate the benchmark. Particularly the prompts under "Preference Dimension Sampler" and "Preference Instantiation" highlight how the preference profiles are task-specific and sparse. The Preference Dimension Sampler identifies which attributes are relevant for a given task-persona pair, and the Preference Instantiation prompt generates specific values and weights only for those relevant attributes.
>
> > "What are the 20 scenarios? What are the relevant and irrelevant attributes for each scenario?"
>
> We refer the reviewer to Section B in the appendix for details on the 20 scenarios. These were randomly selected instances (2 per task across 10 tasks) that we used for human annotation to validate how closely the LLM-generated attributes reflect human judgments. Each scenario includes 10 relevant and 10 irrelevant attributes, generating 400 labels per annotator across 3 annotators, as described in Lines 275-278.
>
> > “How are importance weights calculated?”
>
> The importance weights reflect the relative significance of each preference dimension for a given persona-task pair. We use LLM-based generation because preference importance is inherently context-dependent and subjective, in which what matters most to a user varies based on their background, the task domain, and the specific problem instance. For example, a medical professional studying for an exam may prioritize technical terminology over empathy, while the same person in a personal health crisis would reverse these priorities.
>
> The weights are generated through structured prompting where the LLM considers the persona's background, the task type, and the instantiated preference values to assign "local importance" scores. These scores are then normalized to sum to 1 across all relevant attributes. We refer to the full system prompt in Section F of the appendix under "Preference Instantiation." This approach is validated by our human annotation study (Section B, Lines 275-278), where annotators achieved moderate agreement (Fleiss kappa of 0.463) on attribute relevance—demonstrating that while preference importance involves subjective judgment, there is meaningful consensus that can be captured through principled prompting.
>
> > “How is LLM-based assessment performed?”
>
> A key contribution of our work is developing a rubric-based evaluation framework for personalized reasoning, since it is a fundamentally non-verifiable problem where there is no single “correct” response style. Our approach decomposes preference alignment into constituent attributes, each evaluated independently with explicit criteria and behavioral markers and then combined based on their importance weights. This reduces susceptibility to bias compared to holistic preference judgments, as judges must evaluate against explicit criteria rather than implicit stylistic preferences.
>
> We describe the rubric generation prompts in Appendix F under "Evaluation Rubric Generator." During evaluation, an LLM judge assigns a score for each attribute separately with required justifications. The final preference alignment score is the weighted sum of attribute scores, where weights reflect each dimension's importance to that specific user. This fine-grained approach enables systematic evaluation at scale (10,000 scenarios) while maintaining interpretability through per-attribute justifications.

---

> > ### Author Response · Authors · 2025-11-25
> >
> > **Weakness 2**
> >
> > > "While passive user simulation can simulate more challenging scenarios, it may overlook more informative interactions in general."
> >
> > We appreciate this concern about the realism of passive user simulation. The passive user setting isolates the model's information-seeking capability—testing whether models can proactively identify what information is missing and ask the right questions without relying on user initiative. This represents a challenging but realistic scenario, as research on human-AI interaction documents that users frequently exhibit passive behavior, providing minimal information unless explicitly prompted [1-2].
> >
> > To address the reviewer's concern about overlooking more informative interactions, we are currently running experiments with collaborative user scenarios where users provide richer textual descriptions and proactively reveal preferences without being prompted. We refer the reviewer to Appendix D for the prompts used for collaborative user simulation. These collaborative users represent the other end of the spectrum—more engaged users who volunteer information—allowing us to evaluate whether models can effectively leverage unsolicited preference information. We will report these results in the camera-ready version, providing a more complete picture of model capabilities across different user interaction styles.
> >
> > **Weakness 3**
> >
> > >”"the paper attributes the significant drop in accuracy of the mathematical task to 'over-optimization,' but can this be further demonstrated experimentally?"”
> >
> > The decreasing accuracy with more questions stems from the growing context length and the increasing number of preference constraints the model must satisfy simultaneously. This transforms response generation into a constrained optimization problem: generating the correct answer while respecting user preferences.
> >
> > Most LLMs are trained primarily for answer correctness, with post-training alignment to majority human preferences. However, in reasoning tasks, user preferences often require different reasoning paths to make responses useful for specific individuals. Models trained using reinforcement learning typically collapse onto a single reasoning chain that achieves correctness, becoming brittle at test time when additional user preference constraints are introduced.
> >
> > We point the reviewer to AIME Example 2 in Appendix C.1, which qualitatively demonstrates how this failure case arises. The personalization attempts cause models to "over-optimize" for the extracted preferences but end up sacrificing solution accuracy. This phenomenon is most pronounced in mathematical reasoning (as shown in Figure 5), where formal logical precision is most sensitive to additional contextual constraints.
> >
> > > "Why does the accuracy of the mathematical task continue to decline with increasing question count?"
> >
> > We thank the reviewer for raising this excellent question. We have considered multiple explanations for this phenomenon. Alternative hypotheses include: (1) increasing cognitive load from maintaining multiple preference constraints in working memory, (2) interference between preference-satisfying language patterns and mathematical reasoning patterns, and (3) attention dilution across longer contexts.
> >
> > However, the domain-specific pattern we observe, where mathematical tasks degrade while social reasoning tasks improve with more questions, rules out these general explanations. If the issue were simply context length or cognitive load, we would expect consistent degradation across all task types. Instead, the fact that social reasoning benefits from additional preference information while mathematical reasoning suffers suggests the problem is specific to how formal reasoning interacts with personalization constraints. Mathematical reasoning requires precise logical steps where deviations to satisfy preferences can break the reasoning chain, whereas social reasoning is more robust to stylistic adaptations.
> >
> > [1] Chhabra, S., Kaushal, V., & Girija, S. (2025). Beyond the dichotomy of use and not-use: Forms and motivations of user non-use behaviors toward AI customer service. Computers in Human Behavior.
> >
> > [2] Sundar, S. S., et al. (2020). Rise of machine agency: A framework for studying the psychology of human-AI interaction. Journal of Computer-Mediated Communication, 25(1), 74-88.

---

> > > ### Author Response · Authors · 2025-11-25
> > >
> > > **Weakness 4**
> > >
> > > >”Case studies would be beneficial.”
> > >
> > > We have updated the appendix with various positive and negative examples across different tasks in Appendix C. These case studies illustrate:
> > >
> > > * Successful personalization in social reasoning tasks (Appendix C.2)
> > >
> > > * Failure modes in mathematical reasoning where personalization degrades accuracy (Appendix C.1)
> > >
> > > * Examples of appropriate versus inappropriate preference elicitation strategies
> > >
> > > * Cases where models fail to identify relevant preference dimensions
> > >
> > > These qualitative examples complement our quantitative findings and provide concrete illustrations of the phenomena we observe in aggregate.
> > >
> > > **Weakness 5**
> > >
> > > >”The range of the value of  is inconsistent between Sections 2.3 and 3.”
> > >
> > > Thank you for catching this inconsistency. We have fixed this in the updated manuscript to ensure the range is consistently specified as [1, 5] throughout.
> > >
> > > We hope that our response has answered the reviewer’s questions. If the reviewer’s concerns are resolved, we would appreciate if the reviewer can consider raising the scores!

---

### Official Review · Reviewer_cAyq · 2025-11-01

**Soundness:** 2
**Presentation:** 2
**Contribution:** 2
**Rating:** 6
**Confidence:** 3

**Summary:**

The authors created a benchmark for simulating online exploration of user preferences with a simulator. The authors creates a set of preference attributes and allow the llm to query the value of those attributes' numerical values. They defined the eval metric as the ratio of the improvement performance upon zero shot prompting comparing to oracle where all the preference values are given in context. They argue that personalization is a reasoning capability that current model lacks.

**Strengths:**

1. A novel interactive benchmark for personalization of llm that did not use llm simulate human, which decrease confounding factors.

2. The evaluation spans all frontier models and show that all models failed under the current setting.

**Weaknesses:**

1. While I understand (and even appreciate) the authors’ motivation to control for confounding by simplifying the user simulation with scalar value feedback, the framework is far from realistic human llm interactions. Several components are significantly out-of-distribution: (a) models are explicitly forced to ask preference-related questions via system prompting, and (b) feedback is reduced to scalar or categorical values, whereas real users express preferences through text. These design choices make the evaluation more of a controlled experiment rather than a realistic personalization eval benchmark, though I acknowledge the inherent difficulty of modeling authentic human feedback at scale.
2. Personalization through online interaction and exploration is not a novel topic. Several papers have explored this under llm setting, not even to mention previous bandit literatures. The method authors explored in the previous setting seems to be a soley based on the system prompt, rather than sth like uncertainty quantification. I would wonder if more advanced method can boost the performance.
3. The exact system prompt for exploration is not shown, which I assume would be a very important part of the whole algorithm.

**Questions:**

1. What are the system prompts exactly?

2. Can some more advanced method like UQ based solve this problem better?

3. see weakness 1.

---

> ### Author Response · Authors · 2025-11-25
>
> We thank the reviewer for acknowledging our contributions in creating a novel interactive benchmark for personalization that controls for confounding factors, and for recognizing that our evaluation spans all frontier models and reveals systematic failures across the board.
>
> ## Weakness 1:
>
> > "the framework is far from realistic human llm interactions"
>
> We appreciate this concern and have extended our evaluation to include more realistic user interaction patterns. We now evaluate models on collaborative user scenarios where users provide textual descriptions of preferences and proactively reveal information without being prompted. This captures the case where more vivid textual feedback can be leveraged and serves as a good measure of "steerability" when models are not explicitly instructed to ask questions.
>
> > "models are explicitly forced to ask preference-related questions via system prompting"
>
> We view this as a necessary control to isolate the capability of interest. Without explicit instruction, models default to providing immediate answers without any personalization attempt, making it impossible to evaluate their personalized reasoning capabilities. The prompt simply instructs models to ask questions if needed before responding, which is analogous to how humans naturally gather context before providing advice. The fact that models still fail dramatically even with this instruction demonstrates the fundamental difficulty of the task.
>
> > "feedback is reduced to scalar or categorical values, whereas real users express preferences through text"
>
> We apologize for the misunderstanding, but our passive user actually responds in natural language. We will clarify this in the updated draft. This clean setup follows [1] and can cleanly disentangle the model’s ability to seek information from any other conversational skills.
>
> Furthermore, in response to the reviewer’s concern, we will add numbers for collaborative user scenarios by allowing users to express preferences through more free-form, colloquial conversations and the user is more proactive in providing information.
>
>
> ## Weakness 2:
>
> > "Personalization through online interaction and exploration is not a novel topic. Several papers have explored this under llm setting"
>
> While other benchmarks have explored personalization through online interaction, our benchmark evaluates several novel capabilities as shown in our comparison table (Section 1). Unlike prior work focusing on content recommendation, dialogue consistency, or information-seeking, we are the first to evaluate whether models can (1) identify which preference attributes are relevant for a given task-user pair, (2) strategically elicit sparse preference values through questioning, and (3) adapt their reasoning processes (not just presentation) based on discovered preferences. Furthermore, we consider personalization in the verifiable setting, where the tasks have an objective correct answer, but the reasoning path to the correct answer needs to be adapted to user preferences. In prior personalization work, the end answer changes based on the acquired user preferences.
>
> > "The method authors explored in the previous setting seems to be a soley based on the system prompt, rather than sth like uncertainty quantification. I would wonder if more advanced method can boost the performance."
>
> The whole point and promise of general-purpose AI is that these behaviors should be emergent from training on massive amounts of data. Our benchmark shows exactly the opposite: that even with explicit prompting and simple interaction protocols, current frontier models fail to accomplish personalized reasoning. This establishes the problem as a fundamental capability gap rather than a methods optimization challenge. While more advanced methods like uncertainty quantification may improve performance, our results demonstrate that the basic capability is largely absent, making this a critical research direction.
>
> This is not a methods paper but rather a benchmark and analysis paper that reveals fundamental limitations in current LLMs' personalization capabilities.
>
> ## Weakness 3:
>
> > "The exact system prompt for exploration is not shown"
>
> We have added the complete system prompts to the Appendix section F including prompts for baseline, discovery, and oracle conditions, as well as prompts for user simulation, benchmark generation and evaluation judges.
>
>
> [1] MediQ: Question-Asking LLMs and a Benchmark for Reliable Interactive Clinical Reasoning (Li et al., Neurips 2024)

---

> > ### Comment · Reviewer_cAyq · 2025-11-26
> >
> > Thank you for the detailed response. Some of my earlier questions have been addressed. However, I remain unconvinced by the benchmark setup. I understand the "Without explicit instruction, models default to providing immediate answers without any personalization attempt, making it impossible to evaluate their personalized reasoning capabilities." and your benchmark is trying to show that "While more advanced methods like uncertainty quantification may improve performance, our results demonstrate that the basic capability is largely absent, making this a critical research direction." I believe other interactive personalization benchmarks have a similar goal, and I do not see a clear difference in your setup provides something fundamentally new in terms of realness. I agree that creating a realistic multi turn personalization benchmark is challenging, but the current setup if far from a real daily user - chatbot interaction for personalization. For this reason, I am maintaining my score as a borderline accept.

---

> > > ### Author Response · Authors · 2025-11-26
> > >
> > > We thank the Reviewer for actively engaging with our work, highlighting the “**novel interactive benchmark** for personalization of llm that did not use llm simulate human” and the expansive evaluation aspects of our paper, and acknowledging that our rebuttal has resolved all questions.
> > >
> > > **Novelty**
> > >
> > > We want to clarify the novelty and uniqueness of our work that distinguishes PrefDisco from any existing personalization benchmarks. The Reviewer mentions that "other interactive personalization benchmarks have a similar goal.” We would appreciate it if the Reviewer could provide specific references that they categorize as similar to our work, as this would help us better contextualize our contribution in a constructive manner.
> > >
> > > To clarify the fundamental differences from other personalization benchmarks we have gathered in our extensive literature review, we have added Table 1 (comparison of personalization benchmarks) to the updated draft, which systematically compares PrefDisco against all existing personalization benchmarks across five key dimensions:
> > >
> > > 1. Rubric-based evaluation (instance-specific): PrefDisco is the only benchmark using attribute-specific rubrics rather than generic persona scoring or holistic LLM judgment
> > >
> > > 2. Proactivity in discovering latent preferences: PrefDisco is the only benchmark requiring models to discover sparse, latent preferences through strategic questioning. All other benchmarks either provide preferences upfront or extract them from logs
> > >
> > > 3. Cold-start / just-in-time personalization: PrefDisco is the only benchmark evaluating true cold-start scenarios with no prior interaction history
> > >
> > > 4. Uses verifiable tasks: PrefDisco is the only benchmark combining personalization with tasks that have objective correct answers (reasoning tasks), rather than subjective dialogue or preference-dependent MCQs
> > >
> > > 5. Multi-task breadth: PrefDisco evaluates across 10 diverse reasoning domains
> > >
> > > No existing benchmark targets these capabilities. The comparison table makes this concrete and verifiable. We believe this clearly demonstrates what is "fundamentally new" in our setup beyond what prior work has explored.
> > >
> > > **Realism**
> > >
> > > Regarding "realness," we want to contextualize this within the broader benchmark landscape. Many widely-adopted benchmarks use synthetic setups that are arguably less grounded in real-world scenarios. The value of these benchmarks lies in providing controlled, reproducible measurements of specific capabilities, not in perfectly replicating real-world interactions. Our framework follows this established paradigm while being grounded in psychological research on learner characteristics and validated through human annotation.
> > >
> > > The fact that our proposed interaction paradigm has already been adopted by industry (OpenAI's shopping research feature, released two days ago: https://openai.com/index/chatgpt-shopping-research/) further validates its practical relevance.
> > > This feature adopts precisely our interaction paradigm where the model asks the user some questions first then provides its final response. As an academic work, our contribution lies in establishing this as a measurable research problem with systematic evaluation methodology.
> > >
> > > We hope the comparison table and the industry adoption evidence help clarify the fundamental novelty and practical relevance of our contribution. We would be happy to discuss any specific aspects of the comparison or provide additional clarification. If we have resolved this concern we would appreciate it if the Reviewer would consider raising their score.

---

> > > > ### Comment · Reviewer_cAyq · 2025-11-26
> > > >
> > > > Thank you for your detailed reply. I agree that widely adopted benchmark use synthetic setups, but in my opinion, for personalization benchmark, it's not like swe agent or etc, but a subjective field where we want our setup be as realistic as possible. I do acknowledge the contribution of the paper, which is why I am giving a boarderline accept.

---

### Official Review · Reviewer_CM15 · 2025-11-04

**Soundness:** 3
**Presentation:** 3
**Contribution:** 3
**Rating:** 6
**Confidence:** 4

**Summary:**

This paper proposes PrefDisco, an evaluation methodology that transforms existing static benchmarks into interactive personalization tasks layered on top of the original task. The authors introduce a multi-dimensional numerical profile to instantiate user preferences. Their experiments show that attempts at personalization can hurt performance accuracy over the original task, suggesting naively attempting proactive personalization often degrades alignment for generic responses. Furthermore, they show that models generally do not ask questions, highlighting that interactive personalization remains under-explored.

**Strengths:**

1. Explores a very interesting and relevant problem of multi-turn interactive preference elicitation. Current LLMs tend not to actively discover user preferences, even though this is critical for many real-world problems.
2. The framework is creative and very flexible, allowing researchers to turn any existing benchmark into an interactive personalization task, making it broadly applicable across domains.
3. The evaluation is extensive with various metrics and abundant models and static benchmarks. The counterintuitive finding that personalization attempts can often hurt performance is insightful and valuable to know.

**Weaknesses:**

1. Evaluation with LLM judge: the scoring system relies heavily on an LLM-based judge to score personalization quality. However, the paper does not provide sufficient evidence that the judge’s assessments are reliable or aligned with actual human preferences.  Since the grader model itself embodies particular stylistic biases, it is unclear whether higher scores genuinely reflect improved personalization rather than conformity to the judge model's implicit preferences. It would be helpful to have some sort of evaluation over the reliability of these model-based judges. For example, user studies, human-in-the-loop experiments, calibration checks etc.

2. Persona representation: even though the personas are grounded in psychological research, it remains unclear how well these explicit preference dimensions capture the richness and diversity of real user preferences. User preferences are generally assumed to be high-dimensional, unstructured, and sometimes even internally inconsistent, these modeling dimensions are still fundamentally low order approximations to these preferences.  As a result, models evaluated on this benchmark may be learning to match stylized persona traits rather than genuine personalization applicable to real-world users.

**Questions:**

See weakness above

---

> ### Author Response · Authors · 2025-11-25
>
> We thank the reviewer for acknowledging contributions in proposing the new interesting and relevant problem, highlighting the creativity, flexibility, and the extensive evaluation and findings.
>
> ## Weakness 1
>
> > "Evaluation with LLM judge: the scoring system relies heavily on an LLM-based judge to score personalization quality. However, the paper does not provide sufficient evidence that the judge's assessments are reliable or aligned with actual human preferences."
>
> Unlike prior work where judges simply compare responses or provide holistic scores, we decompose evaluation into individual preference attributes. For each attribute, the judge receives explicit rubric definitions and must provide detailed justifications before assigning scores. This structured approach makes the evaluation process interpretable and reduces susceptibility to implicit stylistic preferences.
>
> We have recruited annotators and are currently conducting human evaluation to validate our LLM judge scores. Our annotation plan involves evaluating a stratified sample of responses across different tasks, models, and preference attributes. Annotators will use the same rubric-based evaluation protocol as the LLM judge, rating responses on individual attributes and providing justifications. We will report inter-annotator agreement and correlation with LLM judge scores in the camera-ready version.
>
> > "Since the grader model itself embodies particular stylistic biases, it is unclear whether higher scores genuinely reflect improved personalization rather than conformity to the judge model's implicit preferences."
>
> We use diverse models  (GPT-4.1, Gemini-2.5-Flash, and Claude-Sonnet-4) randomly selected for each instance during benchmark construction to prevent any single model's biases from dominating. For evaluation, we use GPT-4.1 as a judge throughout. Table 1 provides strong empirical evidence against judge bias. GPT-4.1 scores itself among the lowest across tasks despite being one of the judge models. If stylistic conformity were driving scores, we would expect GPT-4.1 to rate its own outputs highest. The fact that it frequently assigns itself negative scores (worse than baseline) demonstrates that our rubric-based approach successfully evaluates against objective criteria rather than stylistic preferences.
>
> ## Weakness 2
>
> > "Persona representation: even though the personas are grounded in psychological research, it remains unclear how well these explicit preference dimensions capture the richness and diversity of real user preferences."
>
> We acknowledge that real user preferences are complex and high-dimensional. However, our framework is specifically designed to be extensible. The current attribute set was derived from educational psychology research on learner characteristics (Chrysafiadi et al., 2015) and validated through human annotation with moderate inter-annotator agreement (Fleiss kappa of 0.463, 61.5% accuracy against majority voting). Importantly, PREFDISCO's methodology allows researchers to easily incorporate additional attributes or modify existing ones for different domains.
>
> > "User preferences are generally assumed to be high-dimensional, unstructured, and sometimes even internally inconsistent, these modeling dimensions are still fundamentally low order approximations to these preferences."
>
> Our goal is not to claim perfect coverage of all possible preferences, but rather to establish a principled evaluation framework that can systematically measure models' ability to discover and adapt to any specified preference structure. This is a capability that current models demonstrably lack, as our results show. The framework is flexible enough to accommodate any persona generation process, including those with higher-dimensional or more realistic preference distributions.
>
> > "As a result, models evaluated on this benchmark may be learning to match stylized persona traits rather than genuine personalization applicable to real-world users."
>
> We frame this as a constrained optimization problem where models must satisfy discovered preferences while maintaining task correctness. The fact that 29.0% of personalization attempts perform worse than baseline suggests models are not simply pattern-matching to stylized traits, but rather struggling with the fundamental challenge of adapting their reasoning processes to expressed user constraints. This failure mode would not occur if the task were simply about matching surface-level persona characteristics.
>
> ## References
> [1] Konstantina Chrysafiadi, Maria Virvou, et al. Advances in personalized web-based education.
> Springer, 2015

---

> > ### Comment · Reviewer_CM15 · 2025-11-26
> >
> > Thank you for your clarification. I maintain my positive evaluation towards the paper, and I appreciate the authors' additional effort to further conduct human evaluation for LLM judge validation. However, I'm unable to raise my score beyond the current assessment. I understand that the contribution of the paper is framed as providing a flexible and extensible *framework*, but I fundamentally disagree with the claim that evaluating alignment on *specific, predefined preference attributes*, instead of on high-dimensional unstructured latent preference, is a contribution warranting an even higher score than the current one.

---

> > > ### Author Response · Authors · 2025-11-26
> > >
> > > We thank the Reviewer for maintaining a positive evaluation and recognizing important strengths of our work: proposing an interesting and relevant problem, the framework's creativity and flexibility, and the extensive evaluation with insightful findings. We appreciate the constructive engagement throughout the review process.
> > >
> > > **Realism**
> > >
> > > We want to emphasize this work's broader impact. To our knowledge, we are the first to propose and systematically evaluate the paradigm where models proactively ask questions to discover user preferences before providing answers. The real-world relevance is evidenced by OpenAI's recent release (November 24, 2025, after our submission) of their shopping research feature (https://openai.com/index/chatgpt-shopping-research/), which adopts precisely this interaction format. This industry adoption validates that our proposed paradigm addresses genuine practical needs. As an academic work, our contribution lies in establishing this as a measurable research problem with systematic evaluation methodology.
> > >
> > > We also want to contextualize the realism concern within the broader benchmark landscape. Many widely-adopted benchmarks use synthetic setups that are arguably less grounded in real-world scenarios. The value of these benchmarks lies in providing controlled, reproducible measurements of specific capabilities, not in perfectly replicating real-world interactions. Our framework follows this established paradigm while being grounded in psychological research on learner characteristics and validated through human annotation. The fact that our setup has already been adopted by industry further demonstrates its practical relevance.
> > >
> > > **Preference dimensions**
> > >
> > > Regarding preference dimensionality, our framework's contribution is not claiming that predefined attributes are superior to high-dimensional latent preferences, but rather establishing the first systematic methodology for evaluating personalized reasoning in a measurable, reproducible way. The framework is agnostic to the underlying preference representation and could readily accommodate the high-dimensional unstructured latent preference space the reviewer suggests. The key components (interactive preference discovery, sparse instantiation, fine-grained evaluation) remain applicable whether preferences are explicit dimensions or learned latent embeddings.
> > >
> > > Our choice of explicit attributes was motivated by establishing clear ground truth and enabling systematic failure mode analysis. This interpretability allowed us to identify domain-specific brittleness patterns that would be difficult to diagnose in a purely latent space.
> > >
> > > We respectfully note that this work's primary contribution is scientific: formally defining personalized reasoning as a cognitive capability, establishing the first evaluation methodology for measuring it, and revealing systematic failure modes in frontier models. While industry adoption validates the paradigm's real-world relevance, our focus is advancing the research community's understanding of this capability gap and providing tools for systematic measurement.
> > >
> > > We thank the Reviewer for the thoughtful review and for pushing us to clarify the framework's scope, and we hope that this resolves the reviewer’s concerns on the realism of the benchmark.

---

### Author Response · Authors · 2025-12-03
**Discussion Summary to the AC**

We thank the AC for overseeing the review process. Below, we summarize the reviewers’ assessments and how the discussion clarified key concerns.

Our contributions are twofold: 1. We formally define a new capability, **personalized reasoning**, and, 2. We introduce **PREFDISCO**, a benchmark that turns static reasoning benchmarks into interactive, cold-start personalization tasks. Reviewers CM15, cAyq, and 9SvA all recognize the importance of this goal and the core strengths: CM15 notes that we tackle “**a very interesting and relevant problem**” in multi-turn interactive preference elicitation and that our framework is “**creative and very flexible**,” with **extensive evaluation and an interesting counterintuitive result** that naive personalization can harm alignment and accuracy. cAyq highlights that we created a “**novel interactive benchmark** for personalization of llm that did not use llm simulate human,  which decrease confounding factors.” with evaluation spanning all frontier models and showing they fail under the current setting. 9SvA emphasizes that extending from personalized content to personalized reasoning is “**interesting and reasonable**,” that we propose a **psychology-driven evaluation** method, and that our **experiments clearly surface limitations of current LLMs**. Reviewer zi3F highlights that we assessed **21 frontier models across multiple benchmarks**.

The concerns raised (by CM15 and zi3F) are about **LLM-judge reliability** and whether our explicit preference dimensions and personas are “too simplistic” relative to real-world, high-dimensional preferences. In discussion, we clarified that our evaluation is *not* a generic LLM-judge (which we also had described in the main paper): we use a **rubric-based, attribute-level protocol** where, for each preference dimension, the judge receives explicit criteria, must generate a justification, and then assign a score. We also showed that we use **diverse models** (GPT-4.1, Gemini-2.5-Flash, Claude-Sonnet-4) during benchmark construction, and that GPT-4.1 as judge actually scores its *own* outputs among the lowest, which suggests against simple stylistic self-bias. On top of this, we ran a **human evaluation** that mirrors the rubric protocol, where we already see **moderate inter-annotator agreement (Fleiss κ≈0.46)** on attribute relevance. Thus this is in fact an underlying strength: we contribute a **structured, interpretable, and empirically grounded evaluation pipeline** for a fundamentally non-verifiable problem, rather than relying on opaque single-score judgments as in most prior benchmarks.

---

> ### Author Response · Authors · 2025-12-03
>
> The concern that our explicit personas and attributes are “**low-order approximations**” (CM15, 9SvA, zi3F) actually highlights a central design choice and strength of PREFDISCO. We clarified that the current dimensions are **derived from educational psychology** and validated with human annotation, but that the framework itself is **representation-agnostic and extensible**: it can work with explicit attributes, richer domain-specific schemas, or latent preference embeddings. Our primary contribution is to establish the methodology for evaluating whether models can (i) discover which preferences matter in a given task–persona instance, (ii) strategically elicit sparse preference values, and (iii) adapt their reasoning and explanations while preserving correctness. The fact that *models fail badly even in this controlled, explicit setting* is precisely what we wished to highlight: if frontier LLMs cannot solve personalized reasoning where preferences are structured and well-defined, they are even less likely to solve it in messy, high-dimensional real life. In summary this is what makes the benchmark a **clean capability probe** rather than a noisy, underspecified task.
>
> Another concern (from cAyq and zi3F) is **realism of interaction**. We clarified the reviewer’s misunderstanding and stated that (1) the passive user **replies in natural language** (scalars apply only at the hidden preference level), (2) passivity is **motivated by human–AI interaction research** showing users often provide minimal information unless prompted, and (3) system-prompting models to ask questions is a necessary control to isolate the capability we wish to measure. Furthermore, we added a **collaborative user** setting where users proactively provide richer preferences, bracketing a realistic spectrum. More broadly, we are the **first to systematically evaluate the paradigm where models proactively ask questions before providing a personalized answer**, its realistic applicability is shown by OpenAI's adoption of this exact format in their shopping research feature (released November 24, 2025, after our submission). Like widely-adopted benchmarks (AgentClinic, MediQ) that use synthetic setups, our framework provides controlled, reproducible measurements while being grounded in psychological research and validated through human annotation. The reviewers' remarks about artificial setup actually underscore PREFDISCO's strength as a **targeted, controlled stress-test with demonstrated real-world relevance**.
>
> Reviewer zi3F’s main question is that user preference exploration has been studied in a paper they cited titled *UserBench*. In response, we added a **comparison table** and a detailed discussion contrasting PREFDISCO with existing personalization and agent benchmarks (including travel-planning/tool-use settings). We show that PREFDISCO is unique in targeting: (i) **cold-start, just-in-time** personalization with no history; (ii) **strategic discovery of sparse latent preferences via questioning**; (iii) **explanation- and reasoning-level personalization** on tasks with objective correct answers, not just choice of actions; (iv) **rubric-based, instance-specific evaluation** with importance-weighted attributes; and (v) **multi-domain coverage** across 10 reasoning benchmarks. Notably, some “weaker” models (e.g., Claude 3 Opus) outperform stronger ones (e.g., Sonnet-4) on our discovery-mode metrics, reinforcing our core claim: performance gains on static benchmarks do *not* imply gains in just-in-time personalized reasoning. This disconnect between static performance and interactive personalization supported by our large-scale results is precisely the kind of capability gap that we hope our benchmark will prompt future research to address.
>
> During the discussion, we have addressed *all* reviewer concerns and misunderstandings, and provided additional clarification, experiments, qualitative analysis, as well as all requested prompts in the updated draft.
>
> We hope the AC will view PREFDISCO as providing the community with a **structured, extensible way to measure and analyze this missing capability**, so that future work not only improves numbers on existing benchmarks but also makes interactions meaningfully better for real users.

---

### Meta-Review · Area_Chair_1MAb · 2026-01-06

**Summary:**

This paper introduces personalized reasoning as a distinct capability in which LLMs must (i) identify missing user preferences, (ii) strategically elicit them through interaction, and (iii) adapt their reasoning processes accordingly. To study this, the authors propose PREFDISCO, a benchmark that transforms existing static reasoning benchmarks into interactive, cold-start personalization tasks using psychologically grounded personas with sparse, context-dependent preferences. Extensive evaluation across 21 frontier models and 10 tasks reveals systematic failures: naive personalization often degrades alignment, and personalization induces nontrivial accuracy trade-offs, especially in mathematical reasoning. Overall, the paper makes a strong and timely contribution by clearly formulating an underexplored capability gap and providing a scalable, well-structured evaluation framework that exposes limitations of current LLMs.

**Reviewer Concerns:**

The main concerns raised by reviewers focus on (i) the realism and dimensionality of the persona and preference representations, and (ii) reliance on LLM-based judges for evaluation. Reviewers questioned whether explicit, low-dimensional preference attributes sufficiently capture real human preferences, and whether LLM judges reliably reflect human judgments. These concerns were substantially addressed in the rebuttal: the authors clarified that PREFDISCO is a controlled capability probe rather than a full user simulator, emphasized the framework’s extensibility to richer or latent preference representations, and provided additional evidence supporting the robustness of the rubric-based evaluation, including human annotation results and cross-model judge comparisons. Concerns about realism of interaction were also clarified by motivating the passive-user design and by adding a collaborative user setting. No major technical issues remain outstanding.

**Reviewer Scores:**

Reviewers who were initially positive (e.g., CM15, cAyq) maintained their favorable assessments after discussion, though without substantially increasing scores. Reviewers with moderate reservations largely acknowledged that their concerns were addressed through clarification and additional analysis. Overall, reviewers would be expected to retain their current scores, with no strong downward revisions anticipated.

---

### Decision · Program_Chairs · 2026-01-26

Accept (Poster)